# The Association between Salt and Potential Mediators of the Gastric Precancerous Process

**DOI:** 10.3390/cancers11040535

**Published:** 2019-04-15

**Authors:** Susan Thapa, Lori A. Fischbach, Robert Delongchamp, Mohammed F. Faramawi, Mohammed Orloff

**Affiliations:** 1Department of Epidemiology, College of Public Health, University of Arkansas for Medical Sciences, Little Rock, AR 72205, USA; STHAPA@uams.edu (S.T.); RDelongchamp@uams.edu (R.D.); MElfaramawi@uams.edu (M.F.F.); MSOrloff@uams.edu (M.O.); 2Department of Biomedical Informatics, College of Medicine, University of Arkansas for Medical Sciences, Little Rock, AR 72205, USA

**Keywords:** salt, gastric inflammation, epithelial damage, *Helicobacter pylori*

## Abstract

Background: The process by which salt affects the gastric precancerous process has not been adequately studied in humans. Methods: We investigated the effects of salt on gastric inflammation, epithelial damage, the density of *Helicobacter pylori* infection, and gastric epithelial cell proliferation, all of which may be mediators between salt and gastric precancerous/cancerous lesions. These potential mediators were measured using gastric biopsies as: (a) the density of polymorphonuclear and mononuclear cells (gastric inflammation), (b) mucus depletion (gastric epithelial damage), and (c) the severity of *H. pylori* infection. Salt intake was measured with spot urine samples (using urinary sodium/creatinine ratios), self-reported frequency of adding salt to food, and as total added salt. Results: The average sodium/creatinine ratio (at baseline and post-treatment at five months) was associated with increased epithelial damage over the 12-year follow-up period among those with a greater severity of chronic inflammation and among those with continued *H. pylori* infection after treatment at five months. This association was stronger when both severe gastric inflammation and *H. pylori* infection were present at five months (ß: 1.112, 95% CI: 0.377, 1.848). Conclusion: In humans, salt was associated with an increase in epithelial damage in stomachs with more severe previous *H. pylori*-induced chronic inflammation.

## 1. Introduction

Gastric cancer is the third leading cause of cancer deaths worldwide [1,2]. In the development of gastric cancer, gastric inflammation and epithelial damage may occur over many years, and an imbalance between gastric epithelial cell proliferation and apoptosis may facilitate the progression to gastric precancerous lesions and gastric cancer [3,4,5].

There has been a steady decline in the incidence and mortality of gastric cancer since the first half of the 20th century. This decline correlated with the increased use of refrigeration and decreased use of traditional methods of food preservation, such as salting [6]. This led researchers to hypothesize that the decline in the incidence and mortality of gastric cancer may have occurred due to a decreased intake of salt-preserved foods and that salt may be involved in the etiology of gastric cancer [6].

Some epidemiological studies have reported an association between salt intake and advanced lesions (dysplasia or gastric cancer) [7,8,9,10,11,12,13,14,15,16,17,18,19,20,21,22,23,24,25,26]. Most previous cross-sectional, case-control and cohort studies in humans investigated the effects of salt intake on gastric precancerous lesions and/or gastric cancers, but did not examine the underlying mechanism(s) [7,8,9,10,11,12,13,14,15,16,17,18,19,20,21,22,23,24,25,27,28,29,30,31,32,33,34,35,36,37,38,39]. A cross-sectional study by Chen et al. attempted to investigate the mechanism(s) for the effect of salt on the gastric precancerous process by estimating the effects of salt intake on each type of gastric precancerous lesion (atrophic gastritis, intestinal metaplasia and dysplasia) [8], but they did not investigate the effects of salt on gastric inflammation, epithelial damage, the density of *H. pylori* infection and/or gastric epithelial cell proliferation, all of which may be involved in gastric carcinogenesis.

In animal models, there are reports that salt may affect potential mediators (gastric inflammation, epithelial damage, *H. pylori* infection density and gastric epithelial cell proliferation) in the gastric precancerous pathway, which may subsequently increase the risk of gastric cancer [40,41,42]. Evidence from animal studies [40,41] also suggests that *H. pylori* infection may modify the association between salt intake and gastric precancerous lesions and/or gastric cancer [40,41].

Hence, in the current analysis, we explored the associations between salt intake and potential mediators of the gastric precancerous process (gastric inflammation, epithelial damage, density of *H. pylori* infection and gastric epithelial cell proliferation) [3]. 

## 2. Results

In our study, 399 participants provided spot urine samples for the estimation of a sodium/creatinine ratio and 296 participants provided self-reported salt intake measures at baseline. In Appendix A, we show how many participants enrolled, returned for follow-up at each time point, and provided data for each of the study variables, and how many had complete data for each analysis. We investigated the association between salt intake and gastric precancerous progression. In Table 1, we report the linear regression coefficients for the estimated effect of salt intake (measured by urinary sodium/creatinine ratio) on gastric inflammation, epithelial damage and the density of *H. pylori* infection, all of which are thought to affect the association between salt intake and gastric precancerous progression. 

We observed that the sodium/creatinine ratio was associated with a small increase in the change in gastric inflammation over the 12-year follow-up period, with a greater change among those infected with *H. pylori* infection (after treatment) at the five-month follow-up (Table 1). However, the effect on gastric inflammation was not precise. Additionally, for the overall study population, salt intake measured by the sodium/creatinine ratio was associated with a small increase in epithelial damage at 11–12 years compared to after treatment at five months; however, a larger increase in epithelial damage at 11–12 years was observed among those who continued to have *H. pylori* infection after treatment at five months. For every mmol/cg increase in the sodium/creatinine ratio, the epithelial damage score increased by 0.149 among those with *H. pylori* infection at the five-month follow-up assessment (ß: 0.149, 95% CI: 0.007, 0.292) (Table 1).

We further investigated the effect of salt intake on the interactions between (a) gastric inflammation and epithelial damage, and (b) density of *H. pylori* infection and epithelial damage. We observed the sodium/creatinine ratio to be associated with an increase in epithelial damage over the 12-year follow-up period. For every mmol/cg increase in the sodium/creatinine ratio, there was a 0.337 increase in the epithelial damage score in the presence of severe prior gastric inflammation (indicated by the higher density of mononuclear cell infiltrate) at the post-treatment five-month follow-up, particularly among those infected with *H. pylori* (ß: 0.337, CI: 0.048, 0.625) (Table 1). Similarly, the sodium/creatinine ratio was associated with an increase in epithelial damage over the 12-year period; the epithelial damage score increased by 0.489 for every mmol/cg increase in the sodium/creatinine ratio in the presence of a higher density of *H. pylori* infection at the post-treatment five-month follow-up (ß: 0.498, CI: 0.114, 0.881). When both severe gastric inflammation and *H. pylori* infection were present at the post-treatment five-month follow-up, the observed effect of the urinary sodium creatinine ratio on epithelial damage at the 11–12-year follow-up was even stronger (mononuclear cells: ß: 1.112, CI: 0.377, 1.848; polymorphonuclear cells: ß: 1.148, CI: 0.141, 2.155; combined: ß: 1.149, CI: 0.176, 2.122) (Table 1). For every mmol/cg increase in the sodium/creatinine ratio, the epithelial damage score increased by 1.112 when severe chronic gastric inflammation and *H. pylori* infection were present at the five-month follow-up.

Salt intake, measured either as the self-reported frequency of adding salt to food or as the amount of salt added to food, showed observed effects on epithelial damage which were less precise compared to the estimates from urinary sodium/creatinine ratios, but the estimates were in the same direction (Appendix A).

## 3. Discussion

We observed that salt intake measured at the five-month follow-up (after treatment) was associated with increased epithelial damage over the 12-year follow-up period, especially among those with more severe *H. pylori* infection at the five-month post-treatment follow-up and the estimated effect tended to be stronger if the density of *H. pylori* infection was more severe at five months. Furthermore, the presence of prior (five-month) gastric inflammation (mononuclear cells) was also associated with an increase in the estimated effect of salt on epithelial damage at a later stage (11–12 years). The estimated effect of salt intake on epithelial damage at a later stage was even stronger when both severe *H. pylori* infection (at five months) and gastric inflammation (mononuclear cells at five months) were present. These findings suggest that salt intake in the presence of *H. pylori* or gastric inflammation may increase epithelial damage at a later stage and that this association is stronger when both severe *H. pylori* infection and gastric inflammation (mononuclear cells) were previously present. This estimated effect may possibly be due to the potential interaction between salt and *H. pylori* infection to increase gastric carcinogenesis, as previously suggested in a study with Mongolian gerbils [41]. This is also consistent with the findings from our previous analysis [26], where we observed an association between salt intake and advanced stage gastric precancerous lesions or cancer at the 11–12-year follow-up, especially among those with prior (five-month) *H. pylori* infection. Salt intake may increase epithelial damage at a later stage, particularly in the presence of severe *H. pylori*-induced gastric inflammation, which may then increase the progression to advanced gastric precancerous lesions (dysplasia) or cancer. 

Associations between salt intake during the first five months of follow-up and the outcomes (gastric inflammation, the density of *H. pylori* infection and epithelial damage) were observed in our study with 11–12-year follow-up, which is also consistent with the results of previous animal studies [40,41,43]. In studies of animals sustained on a high-salt diet, salt intake was associated with a higher density of total inflammatory cell infiltrate (mononuclear and polymorphonuclear cells) [41], a higher density of *H. pylori* infection [40], and an increase in mucosal damage and bleeding [40,43]. In our study, salt intake during the first five months was observed to be associated with an increase in the density of polymorphonuclear and/or mononuclear cells, the density of *H. pylori* infection, and epithelial damage in humans over time, although these associations were not precise. 

An effect of salt on the density of *H. pylori* infection, gastric inflammation, and epithelial damage may also occur as a result of the increased proliferation of gastric epithelial cells [44]. Associations between salt and ornithine decarboxylase activity (ODC) activity have been reported in animal studies [40,42,44]. ODC is a key enzyme in the synthesis of polyamines from arginine to ornithine and has been established as the measure of cellular proliferation [45]. The increased proliferation of gastric epithelial cells may provide more space for *H. pylori* colonization, thereby increasing the density of *H. pylori* infection [21,41], leading to gastric inflammation and epithelial damage, and subsequent increase in the risk of gastric precancerous lesions and/or gastric cancer [3,46]. In a small subset of 56 participants in our cohort, gastric epithelial cell proliferation was estimated using biopsies from baseline and the five-month follow-up. Gastric epithelial cell proliferation was measured as the activity of ODC [45]. In this subset of participants, we observed a weak estimated effect of salt on gastric epithelial cell proliferation. For every one mole/cg increase in the average sodium/creatinine ratio at baseline and five months, we observed an increase in ODC activity at baseline by 0.010 pmoles, ODC activity at five months by 0.017 pmoles, and change in ODC activity from baseline to five months by 0.016 pmoles. However, we did not have 11–12-year follow-up data for ODC activity and so ODC was not a primary outcome of interest in the current analysis.

Our study has some limitations. First, the misclassification of study outcomes may have occurred because the measures of gastric inflammatory cells, epithelial damage and density of *H. pylori* infection were subjective, based on the pathologists’ expert judgement. This misclassification is likely non-differential because the pathologists were unaware of the status of salt intake in the study participants at the time of the assessment of the outcomes. If the non-differential misclassification was independent of other variable classification errors, it would be expected to result in an underestimation of the magnitude of the association and would not be expected to explain away the observed association between salt intake and the outcomes. In addition, we attempted to reduce the misclassification by having three study pathologists who provided independent assessments of the outcomes and reached a consensus in the case of any discrepancies. Second, multiple follow-up assessments are ideal to account for changes in exposure and confounders over time; however, only two follow-up assessments (at five months and at 11–12 years) were done in the original cohort study. Third, we measured the outcomes (potential mediators) using a standardized scoring method (0 (none)–3 (severe)) without objective units. Finally, salt intake was measured as the urinary sodium/creatinine ratio from spot urine samples and self-reported dietary intake. Self-reported measures may not accurately reflect the total intake of salt as the participants may not be aware of the amount of salt added while cooking or naturally present in the food items. Also, sodium/creatinine ratios may not reflect the life-long intake of salt. However, in the Colombian population, individual dietary habits may be more consistent over time than in developed countries. Urinary sodium/creatinine ratios have been reported by multiple studies to be more accurate in measuring salt intake than self-reported measures because participants may not be aware of the amount of salt present in the foods they consume [47,48,49,50]. A non-differential misclassification of self-reported salt intake measure is likely because the histological outcomes at follow-up or the change in these histological outcomes from baseline to follow-up would not have occurred at the time participants were interviewed for salt intake measures. Compared to the estimates of effect obtained from salt intake from the urinary sodium/creatinine ratio, the estimates obtained from salt intake measured as the frequency of adding salt to foods and the total added salt were less precise, but in the same direction. 

Despite these limitations, our study has several strengths. First, we used multiple approaches to measure salt intake because each approach may have its own limitations, and included an objective measure which does not rely on self-reports. Second, previous studies estimating the effects of salt intake on gastric inflammation, epithelial damage, the density of *H. pylori* infection and gastric epithelial cell proliferation were conducted in animals; here, we estimated the effects in humans. Finally, this was a cohort study with long-term follow-up, which enabled us to study changes over adequate time to estimate the effects of salt intake on histological changes. 

The outcomes in this study are potential mediators in the pathway between salt intake and gastric precancerous lesions/gastric cancer. Future studies should conduct formal mediation analyses using methods developed by Van der Weele [51] in cohorts with the exposure (salt intake), the mediators (gastric inflammation, epithelial damage, density of *H. pylori* infection and gastric epithelial cell proliferation), and the outcomes (gastric precancerous lesions/cancer) to evaluate and quantify the potential pathways of gastric carcinogenesis followed over adequate time [46]. It should be noted that such mediation analyses would require repeated follow-up of a much larger cohort at high risk for advanced precancerous lesions and gastric cancer. 

## 4. Materials and Methods

### 4.1. Study Population

In the current paper, we analyzed data from a prospective cohort study conducted in Pasto, Colombia, that consisted of a 16-week randomized clinical trial in which study participants were randomly assigned to one of the following five treatment regimens: (a) metronidazole, amoxicillin and bismuth subsalicylate for the first two weeks and bismuth subsalicylate alone for the next 14 weeks; (b) calcium carbonate (weeks 1–16); or (c) treatment regimens (a) and (b), (d) tetracycline (weeks 1–16), or (e) placebo (weeks 1–16) [52]. The participants were evaluated at five months (14 weeks after treatment) and 11–12 years after initiating the treatments. Most participants were highly motivated as they were primarily recruited through a public service radio announcement and typically had to stand in line to be screened for entrance into the cohort [52]. At baseline, we collected detailed contact information from the participants and their friends and family, who could always locate them to assist us with long-term follow-up of the participants. We included individuals who were between 18 and 65 years old, had symptoms consistent with non-ulcer dyspepsia, planned to reside within the Pasto city limits for at least five years, agreed to provide informed consent, and were otherwise healthy [52]. Participants were not eligible if they took medications or had conditions that could interfere with the trial medications (pregnancy, allergies to trial medications, etc.). Those who did not have baseline *H. pylori* infection or who had baseline dysplasia, gastric cancer, or ulcers were excluded from the clinical trial, but not from the long-term follow-up assessment (11 to 12 years post-baseline) [52]. 

### 4.2. Measurement of Dietary Salt Intake 

Salt intake evaluated using dietary recall is problematic as salt intake in food varies and most people do not know the amount of salt contained in their food. Therefore, we focused on a more objective measure using sodium creatinine ratios in addition to other measures of salt intake. We measured salt intake in the following three ways: (1) the ratio of urinary sodium/creatinine, (2) self-reported frequency of adding salt to food, and (3) collecting salt added to food. At baseline, five-month follow-up post-treatment, and 11–12-year follow-up, urine samples from the first void of the day were obtained from all participants during the visit after an overnight fast. Sodium was determined by flame photometry, while creatinine was measured by the Jaffe reaction [8]. We estimated urinary sodium/creatinine ratios from the measured concentrations of sodium (mmol/L) and creatinine (mg/dL) by dividing urinary sodium in mmol/L by urinary creatinine in g/L. In the analysis, we converted sodium/creatinine ratios from mmol/g to mmol/cg for meaningful interpretations. Urinary sodium levels were corrected for creatinine levels to account for urinary volume and dilution. To increase the reliability and validity of our study, we used the average of the sodium/creatinine ratios at baseline and five months [47].

At baseline, research personnel interviewed the participants regarding the frequency with which the study participant, or any individual who cooked in their household, added salt to food; the self-reported frequency was categorized into the following groups: rarely or never adds salt to foods, occasionally adds salt to foods, and always or frequently adds salt to foods. Lastly, we estimated total salt added to food by asking the participants to add the same quantity of salt added to food to a container. At follow-up visits, the amount of salt in the container was measured for analysis.

### 4.3. Measures of Gastric Inflammation, Epithelial Damage and H. pylori Infection

*H. pylori* infection, gastric inflammation and epithelial damage were measured from biopsies for the entire cohort at each time point [52]. The biopsies collected at baseline, five months, and 11–12 years were embedded in paraffin, and sectioned and stained with hematoxylin-eosin [52]. For each participant, biopsies were taken from the antrum and corpus of the stomach. For each biopsy, three pathologists provided independent histological reports, and any discrepancies were resolved through discussions among the pathologists until a consensus was reached [52]. These histologic reports included data regarding gastric inflammation, epithelial damage and the density of *H. pylori* infection measured from an average of four biopsies at baseline, four biopsies at the five-month follow-up, and six biopsies at the 11–12-year follow-up. We measured gastric inflammation as the average and maximum (a) density of the polymorphonuclear leukocyte infiltrate, (b) density of the mononuclear leukocyte infiltrate, and (c) combined density of the polymorphonuclear leukocyte infiltrate and the mononuclear leukocyte infiltrate across biopsies. We measured gastric epithelial damage as the average and maximum mucous depletion measured across biopsies. The study pathologists measured the density of *H. pylori* infection using a modified Steiner stain [52]. For each biopsy, the study pathologists scored gastric inflammation, epithelial damage and the density of *H. pylori* infection from 0–3, where 0 represented the lowest density of gastric inflammatory cell infiltrate, the lowest density of *H. pylori* infection and the lowest degree of epithelial damage (i.e., depletion of gastric mucus).

### 4.4. Measurement and Assessment of Potential Confounders

We identified potential confounders by creating Directed Acyclic Graphs [53] based on the existing literature. Directed Acyclic Graphs depict associations between exposures, outcomes and covariates using existing evidence [53]. In the current analysis, the outcomes of interest were the mediators for the association between salt intake and gastric precancerous progression (gastric inflammation, epithelial damage and *H. pylori* density) and not the progression itself. However, since the literature lacks information regarding factors which may confound the association between salt intake and each potential mediator, we adjusted for potential confounders (identified based on the existing literature) for the association between salt intake and gastric precancerous progression. We identified socioeconomic status [54], fruit and vegetable intake [54] and age [54] as potential confounders. In Colombia, car ownership at baseline was generally limited to people of a higher socio-economic status; hence, we used car ownership collected from the baseline interviews as an estimate of socioeconomic status. Fresh fruits and vegetables intake, another potential confounder, was measured as the total number of servings of fresh fruit (including juices) and vegetables per week using questionnaires, and the ages of the participants were collected from their Colombian identification cards.

### 4.5. Statistical Analysis

Our objective was to estimate the effects of salt intake (urinary sodium/creatinine ratios, reported frequency of adding salt to food, and estimated salt intake) on the following: (a) the average and maximum gastric inflammation, epithelial damage and *H. pylori* infection density across biopsies; (b) the change in the average and maximum gastric inflammation, epithelial damage and *H. pylori* density infection across biopsies from baseline to the 11–12 year follow-up; and (c) how epithelial damage would be influenced by prior gastric inflammation and/or density of *H. pylori* infection across biopsies (two-way and three-way interactions), since both gastric inflammation and epithelial damage may be necessary for progression in the gastric precancerous process. To achieve the above-mentioned objectives, we conducted linear regression analyses, adjusting for the following potential confounders identified by Directed Acyclic Graphs [53]: age, socio-economic status and fresh fruit and vegetable intake. We also adjusted for the baseline measures when the outcome was the change in a measure over time. We repeated all analyses (except for density of *H. pylori* infection as the outcome) among those who did not have *H. pylori* infection at the five-month post-treatment follow-up, those who continued to be infected with *H. pylori* at the five-month post-treatment follow-up, and those who were persistently infected with *H. pylori* at both the five-month post-treatment and the 11–12-year follow-up. This was done to examine if *H. pylori* infection status affected the associations between salt intake and the outcomes (gastric inflammation and epithelial damage). For each analysis, we estimated beta-coefficients, which in our study, would estimate the effect of a one-unit increase in salt intake on the outcomes. For example, for the estimated effect of salt intake (using sodium/creatinine ratios measured in mmol/cg) on chronic inflammation (measured on a scale from 0 or no mononuclear leukocyte infiltrate to 3 or severe mononuclear leukocyte infiltrate), the beta-coefficient can be interpreted as an estimate of the change in the chronic inflammation score per mmol/cg of the sodium/creatinine ratio. We used 95% confidence intervals to estimate the precision of these estimates. Consistent with recommendations by the American Statistical Association, our analysis focused on effect estimation rather than statistical significance testing as the appropriate analytic goal [55,56].

## 5. Conclusions

Understanding how salt affects gastric carcinogenesis may be helpful in public health efforts to prevent gastric cancer. Our current results provide preliminary evidence in humans that salt intake may lead to an increase in epithelial damage, especially among those with earlier gastric inflammation and/or *H. pylori* infection. Epidemiologic cohort studies with larger sample sizes are needed to further understand the mechanisms by which dietary salt may contribute to the development of gastric cancer. 

## Figures and Tables

**Table 1 cancers-11-00535-t001:** Results of linear regression for the estimated effect of salt intake measured as the average of baseline and five-month measures of the urinary sodium/creatinine ratio at baseline on gastric inflammation and epithelial damage ^a^.

Outcome		Level	Adjusted ß (95% Confidence Intervals) ^b^			
			Overall (*n* = 259)	*H. pylori* Positive at 5 Months (*n* = 215)	*H. pylori* Negative at 5 Months (*n* = 44)	Persistent, *H. pylori* Infection (Positive at 5 Months and at 11–12 Years) (*n* = 186)
Gastric Inflammation	Change in average inflammation (5 months vs. 11–12 years)	PMN	0.049 (−0.088, 0.187)	0.052 (−0.092, 0.196)	0.109 (−0.219, 0.437)	0.000 (−0.118, 0.118)
		Mononuclear	0.004 (−0.072, 0.080)	0.026 (−0.050, 0.101)	−0.008 (−0.257, 0.241)	−0.013 (−0.071, 0.046)
		Combined	0.028 (−0.069, 0.126)	0.040 (−0.062, 0.142)	0.056 (−0.218, 0.329)	−0.006 (−0.084, 0.072)
	Change in maximum inflammation (5 months vs. 11–12 years)	PMN	0.089 (−0.067, 0.246)	0.085 (−0.072, 0.242)	0.198 (−0.185, 0.580)	0.033 (−0.094, 0.161)
		Mononuclear	0.018 (−0.085, 0.120)	0.041 (−0.072, 0.153)	−0.011 (−0.264, 0.243)	0.012 (−0.085, 0.110)
		Combined	0.069 (−0.047, 0.184)	0.079 (−0.044, 0.202)	0.098 (−0.188, 0.384)	0.029 (−0.073, 0.131)
Epithelial Damage	Change in average Epithelial damage (5 vs. 11–12 years)		0.118 (−0.018, 0.255)	0.149 (0.007, 0.292)	0.103 (−0.285, 0.490)	0.108 (−0.017, 0.234)
	Change in maximum Epithelial Damage (5 vs. 11–12 years)		0.105 (−0.055, 0.264)	0.122 (−0.044, 0.289)	0.168 (−0.290, 0.627)	0.063 (−0.072, 0.199)
Interaction between gastric inflammation and epithelial damage	Inflammation at 5 months * Mucus Depletion at 11–12 years	PMN	0.134 (−0.270, 0.539)	0.313 (−0.114, 0.741)	−0.275 (−0.720, 0.169)	0.214 (−0.195, 0.624)
		Mononuclear	0.231 (−0.030, 0.492)	0.337 (0.048, 0.625)	0.020 (−0.464, 0.504)	0.238 (−0.024, 0.500)
		Combined	0.154 (−0.208, 0.515)	0.304 (−0.092, 0.701)	−0.136 (−0.723, 0.450)	0.186 (−0.172, 0.544)
	Inflammation at 11–12 years * Mucus Depletion at 5 months	PMN	0.068 (−0.271, 0.406)	0.182 (−0.186, 0.549)	−0.081 (−0.500, 0.339)	0.094 (−0.271, 0.458)
		Mononuclear	0.066 (−0.221, 0.354)	0.170 (−0.140, 0.480)	−0.131 (−0.503, 0.241)	0.111 (−0.209, 0.431)
		Combined	0.045 (−0.283, 0.373)	0.157 (−0.195, 0.508)	−0.140 (−0.592, 0.312)	0.083 (−0.278, 0.443)
*H. pylori* infection density	Change in average *H. pylori* density (5 months vs. 11–12 years)		0.032 (−0.117, 0.181)	0.058 (−0.091, 0.207)		0.039 (−0.093, 0.172)
	Change in max *H. pylori* density (5 months vs. 11–12 years)		−0.010 (−0.184, 0.164)	0.013 (−0.165, 0.191)		-0.021 (−0.154, 0.112)
Interaction between density of *H. pylori* infection, inflammation and epithelial damage	Density of *H. pylori* infection at 5 months * Mucus Depletion at 5 months		0.120 (−0.157, 0.397)	0.191 (−0.099, 0.481)		0.175 (−0.130, 0.480)
	Density of *H. pylori* infection at 11–12 years * Mucus Depletion at 11–12 years		0.185 (−0.262, 0.633)	0.355 (−0.123, 0.834)		0.311 (−0.067, 0.690)
	Density of *H. pylori* infection at 5 months * Mucus Depletion at 11–12 years		0.322 (−0.064, 0.708)	0.498 (0.114, 0.881)		0.373 (0.032, 0.714)
	Density of *H. pylori* infection at 11–12 years * Mucus Depletion at 5 months		0.041 (−0.284, 0.365)	0.119 (−0.239, 0.478)		0.069 (−0.273, 0.412)
	Density of *H. pylori* infection at 5 months * Inflammation at 5 months * Mucus Depletion at 11–12 years	PMN	0.766 (−0.167, 1.699)	1.148 (0.141, 2.155)		0.940 (−0.058, 1.938)
		Mononuclear	0.774 (0.083, 1.464)	1.112 (0.377, 1.848)		0.907 (0.182, 1.633)
		Combined	0.756 (−0.163, 1.676)	1.149 (0.176, 2.122)		0.906 (−0.038, 1.849)
	Density of *H. pylori* infection at 11–12 years * Inflammation at 11–12 years * Mucus Depletion at 5 months	PMN	0.100 (−0.877, 1.077)	0.373 (−0.714, 1.460)		0.238 (−0.830, 1.307)
		Mononuclear	0.230 (−0.609, 1.070)	0.491 (−0.442, 1.425)		0.388 (−0.524, 1.300)
		Combined	0.106 (−0.872, 1.083)	0.377 (−0.707, 1.462)		0.239 (−0.819, 1.297)

^a^ The change in score for the outcome per mmol/cg of the sodium/creatinine ratio. * Adjusted for age, car ownership and fresh fruit and vegetable intake; for changes in gastric inflammation, epithelial damage and the density of *H. pylori* infection, we also adjusted for baseline measures.

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
