# Peer review of "The Association between Salt and Potential Mediators of the Gastric Precancerous Process"

_cancers, 2019, doi:10.3390/cancers11040535_

Round 1
Reviewer 1 Report
In this manuscript, the authors are trying to evaluate the association The association between salt and other potential factors causing gastric precancerous process. However, there are multiple issues remained to be addressed:
1. The language in this manuscript should be more concise. Tedious sentences would exhaust readers patience;
2. There should be a representative figure for the study plan described in section 2.1;
3. How many cases of biopsy in this study? It is better to show some representative figures based on their score system;
4. How did the sodium/creatinine ratio, salt intake and biopsy link to each other? Did they belong to totally different participants? Or some of the info. belongs to a same participant? If the data were collected from different individuals, it might be more convincing to compare one single factor while pairing other similar factors into one category.
5. Please use some bar charts or scatter plots or any other visual figure to present the data, they are a lot better than just tables;
6. Why did the authors choose 5 months and 11-12 years as the time points for investigation? Any evidence? Would it be better if they set several periods of time and investigate the correlation coefficients of the factors within these time point?
7. Any outliers in this study? How did the authors deal with them?
8. It might be better to show the medians and P values and some other relevant parameters instead of only linear regression coefficients in the table.
Author Response
We would like to thank the reviewers for their comments and suggested edits. Below are the reviewers’ comments (in quotation marks) followed by our point-by-point response (in italics).
Reviewer: “In this manuscript, the authors are trying to evaluate the association The association between salt and other potential factors causing gastric precancerous process. However, there are multiple issues remained to be addressed:
1. The language in this manuscript should be more concise. Tedious sentences would exhaust readers patience;”
Response: We are not sure which sentences Reviewer 1 is referring to. However, we have rephrased specific sentences according to comments made by Reviewer 1 and the other reviewers (as described in our subsequent responses below). We hope that these are the sentences that Reviewer 1 is referring to.
Reviewer: “2. There should be a representative figure for the study plan described in section 2.1;”
Response: We have added Figure S1 and Figure S2 with flow charts which represents the study plan described in section 4.1. (Please note that the publishers require the Material and Methods section to follow the Discussion section; therefore, our previous section 2.1 is now 4.1).
Reviewer: 3. “How many cases of biopsy in this study? It is better to show some representative figures based on their score system;”
Response: For the current analyses, we used all biopsies which were evaluated by the pathologists with regard to the histological variables of interest in this study. The histological reports with these variables of interest came from an average of 4 biopsies at baseline, 4 biopsies at the 5-month follow-up and 6 biopsies at the 11-12-year follow-up. We have modified sentences under section 4.3 to clarify this.
Reviewer: “4. How did the sodium/creatinine ratio, salt intake and biopsy link to each other? Did they belong to totally different participants? Or some of the info. belongs to a same participant? If the data were collected from different individuals, it might be more convincing to compare one single factor while pairing other similar factors into one category.”
Response: Sodium creatinine ratios, salt intake and biopsies were collected from the same cohort of individuals that were followed up from baseline to 11-12 years. The first sentence of section 4.3 states, “H. pylori infection, gastric inflammation, and epithelial damage were measured from biopsies for the entire cohort at each time point [51]” to clarify that these variables were collected from the entire cohort. We hope that the addition of the flow charts in Figure S1 and Figure S2 also clarifies this.
Reviewer: “5. Please use some bar charts or scatter plots or any other visual figure to present the data, they are a lot better than just tables;”
Response: We are using the beta-coefficients to estimate measures of effect from linear regression analyses adjusted for potential confounders. We are confused as to how Reviewer 1 would like us to represent these estimates in a scatter plot or other visual figure.
Reviewer: “6. Why did the authors choose 5 months and 11-12 years as the time points for investigation? Any evidence? Would it be better if they set several periods of time and investigate the correlation coefficients of the factors within these time point?”
Response: Although more frequent follow-up would have been preferred, funding limitations prevented more frequent follow-up. Additionally, the original cohort study was conducted to evaluate the short- and long-term anti-inflammatory and tissue protectant effects of the trial medications. Short-term evaluation was done at 5 months and the long-term follow-up was done at 11-12 years to evaluate the long-term effects. In the original cohort (from 1993-2005) we did not conduct follow-up assessments at additional time points. We now mention this as a limitation in the 4th paragraph of the discussion section.
Reviewer: “7. Any outliers in this study? How did the authors deal with them?”
Response: We did not observe extreme outliers in our data, and all data used were within valid ranges.
Reviewer: “8. It might be better to show the medians and P values and some other relevant parameters instead of only linear regression coefficients in the table.”
Response: We do not know which medians and p-values Reviewer 1 would like us to present, nor how a median value or a p-value would help with the interpretation of the results consistent with our objectives. We strongly agree with Reviewer 3 that the analysis should not be based on p-values and hypothesis testing. Perhaps our response to Reviewer 3’s comments regarding our interpretation of the beta-coefficients as measures of effect will make the results clearer for the readers. Further, (consistent with Reviewer 3’s comments) we are following recommendations from the American Statistical Association and leading methodologists in epidemiology. Perhaps Reviewer 1 would find it helpful to read the American Statistical Association’s recommendation regarding p-values (Ronald L. Wasserstein & Nicole A. Lazar (2016) The ASA's Statement on p-Values: Context, Process, and Purpose, The American Statistician, 70:2, 129-133, DOI: 10.1080/00031305.2016.1154108). Highlighting the rarity and importance of this statement, this educational policy statement was one of the few made by the ASA over the last 150 years. As pointed out by the American Statistical Association in their Statement, “let us be clear. Nothing in the ASA statement is new. Statisticians and others have been sounding the alarm about these matters for decades, to little avail. We hoped that a statement from the world’s largest professional association of statisticians would open a fresh discussion and draw renewed and vigorous attention to changing the practice of science with regards to the use of statistical inference.”
Reviewer 2 Report
The manuscript is quite well written.
The methods are adequate.
The results justify the conclusions drawn.
Author Response
Reviewer: “The manuscript is quite well written. The methods are adequate. The results justify the conclusions drawn.”
Response: We are glad you liked our manuscript!
Reviewer 3 Report
This paper presents the results of well-designed and well-executed epidemiologic research on a topic of major public health significance.
I have only minor changes to suggest:
General
--Check for typos (I spotted a few instances of missing punctuation, missing words or repeated words): for example, lines 24, 107, 113, 140, 142, 151, 152, 206 , 220
--Citation numbering jumps from 42 to 51
Abstract
--Conclusion: temper the cause-effect claim: instead of “…salt appears to increase…damage…” use wording such as “in this study, salt was associated with an increase…”.
Methods
--Line 113: reword “…density measured with…stain” to avoid the impression that the stain does the measurement instead of the pathologist
--Line 114: specify who did the scoring
--Lines 122-25: this sentence is hard to unpack because the distinction between gastric cancer progression and mediators is subtle; I had to read it several times before I realized it was not a contradictory statement; I suggest inserting a preceding sentence that emphasizes the distinction between the mediators as outcomes and progression as an outcome
--Briefly describe methods used to maximize follow-up and completeness of collected data
Statistical Analysis
--Specify explicitly the measures of effect and precision estimated in the analyses and their interpretation with respect to the analysis goals
--Lines 151-152: move these lines to Results
Results
--Add a table or chart that shows how many participants enrolled, returned for follow-up at each time point, and provided data for each of the study variables - and how many had complete data for each analysis
--Interpret the magnitude of estimated effects in terms of the beta-coefficient units, as you do in lines 239-241 to describe the results of another study
--Line 169: refer to precision rather than statistical significance, given that your analysis is not based on significance testing (nor should it be)
--Line 176: Clarify the meaning of “tended to” (I suggest avoiding this phrase, which does not have a precise statistical meaning)
--Reword the following lines to avoid cause-effect claims such as “…sodium/creatinine ratio… increase…damage”: 176, 180
--Use “observed effect(s)” or “estimated effect(s) instead of “effect(s)” in the following lines: 183, 187
Tables
--Indicate units for the beta-coefficients
Discussion
--Reword the following lines to avoid cause-effect claims: 195, 225
--Use “observed effect(s)” or “estimated effect(s) instead of “effect(s)” in the following lines: 193, 195
-- Clarify the meaning of “tended to” (I suggest avoiding this phrase, which does not have a precise statistical meaning), in the following lines: 193, 224
--Line 227: refer to precision rather than statistical significance, given that your analysis is not based on significance testing (nor should it be)
--Line 230: If this is the first use of ODC, spell it out and describe what it is here rather than later in paragraph
--Line 231: “colonization of H. pylori” should be “H. pylori colonization” or “colonization by H. pylori”
--Line 246: I suggest changing “opinion” to “expert judgment”
--Lines 248-9: given that predictions about “bias” are generally relevant only to expected results, I suggest rewording this sentence to something like “If the non-differential misclassification was independent of other variable classification errors, it would be expected to result in underestimation of the magnitude of the association.”
--Lines 252-264: I suggest dropping the discussion of multiple hypothesis testing given the following: the description of statistical methods does not refer to statistical hypothesis testing as a goal; the analysis appropriately presents confidence intervals rather than p-values; the use of statistical significance testing and Bonferroni corrections have long been considered poor practice in epidemiology. Instead, I suggest restricting any justifications of the statistical methods used to the Methods section, where, as noted above, I suggest that you explicitly describe and justify the measures used for effect estimation. In that section, you can reference epidemiologic methods literature that explains why effect estimation, rather than statistical significance testing is the appropriate analytic goal. If you feel it’s necessary, you can also reference literature that explains why the Bonferroni correction is not appropriate.
Conclusion
Line 293: I suggest removing the first sentence of the conclusion, which is overly vague.
Lines 298-300: Clarify the meaning of the last sentence of the conclusion; as written; it sounds like it is a foregone conclusion that salt will prevent progression of gastric cancer progression, that doing so is of relevance to all patients, that clinicians need help to advise patients to reduce salt intake, and that understanding mechanisms will help clinicians advise their patients, none of which can be assumed to be true.
Author Response
We would like to thank the reviewers for their comments and suggested edits. Below are the reviewers’ comments (in quotation marks) followed by our point-by-point response (in italics).
Reviewer: “This paper presents the results of well-designed and well-executed epidemiologic research on a topic of major public health significance. I have only minor changes to suggest: General: Check for typos (I spotted a few instances of missing punctuation, missing words or repeated words): for example, lines 24, 107, 113, 140, 142, 151, 152, 206 , 220.”
Response: We have gone through the manuscripts and made edits as suggested for these lines. We hope that the line numbers we modified match the line numbers the reviewer is referring to. (Note: the publisher requires us to place the Material and Methods section after the Discussion section, so the previous line numbers differ from those seen in the previous version.)
Reviewer: “--Citation numbering jumps from 42 to 51.”
Response: We were confused with the publishers’ guidelines of placing the Material and Methods section after the Discussion section. We now have the sections in the order specified by the publishers and the references should now be in order.
Reviewer: “Abstract--Conclusion: temper the cause-effect claim: instead of “…salt appears to increase…damage…” use wording such as “in this study, salt was associated with an increase…’”
Response: We have reworded the conclusions of the abstract as suggested by Reviewer 3.
Reviewer: “Methods--Line 113: reword ‘…density measured with…stain’ to avoid the impression that the stain does the measurement instead of the pathologist.”
Response: We have replaced this sentence as follows: “The study pathologists measured the density of H. pylori infection using a modified Steiner stain.”
Reviewer: “Line 114: specify who did the scoring.”
Response: We have reworded this sentence as follows: “For each biopsy, the study pathologists scored gastric inflammation, epithelial damage, and the density of H. pylori infection from 0-3...”
Reviewer—“Lines 122-25: this sentence is hard to unpack because the distinction between gastric cancer progression and mediators is subtle; I had to read it several times before I realized it was not a contradictory statement; I suggest inserting a preceding sentence that emphasizes the distinction between the mediators as outcomes and progression as an outcome.”
Response: We have added a sentence and reworded the relevant sentence as follows: “In the current analysis, the outcomes of interest were the mediators for the association between salt intake and gastric precancerous progression (gastric inflammation, epithelial damage and H pylori density) and not the progression itself. However, since the literature lacks information regarding factors which may confound the association between salt intake and each potential mediator, we adjusted for potential confounders (identified based on the existing literature) for the association between salt intake and gastric precancerous progression...”
Reviewer: “--Briefly describe methods used to maximize follow-up and completeness of collected data.”
Response: We have added information to the first paragraph of the Material and Methods section to briefly describe the methods for enrolling participants to maximize follow-up and completeness of data.
Reviewer: “Statistical Analysis--Specify explicitly the measures of effect and precision estimated in the analyses and their interpretation with respect to the analysis goals.”
Response: We have added the following sentence to the Statistical Analysis section: “For each analysis, we used the beta coefficient from a linear model to estimate the effect of a one-unit increase in sodium-creatinine on the outcome score, and the precision was estimated using a 95% Confidence Interval”. We have also included sentences in the results section which interpret the beta coefficient accordingly.
Reviewer: “--Lines 151-152: move these lines to Results.”
Response: This sentence has now been moved to the Results section.
Reviewer: “Results --Add a table or chart that shows how many participants enrolled, returned for follow-up at each time point, and provided data for each of the study variables - and how many had complete data for each analysis.”
Response: We have added flowcharts (Figure S1 and Figure S2) to show how many participants enrolled, returned for follow-up at each time point, and provided data for each of the study variables - and how many had complete data for each analysis.
Reviewer: “--Interpret the magnitude of estimated effects in terms of the beta-coefficient units, as you do in lines 239-241 to describe the results of another study.”
Response: We now interpret the magnitude of the estimated effects in terms of the beta-coefficient units in the Results section.
Reviewer: “--Line 169: refer to precision rather than statistical significance, given that your analysis is not based on significance testing (nor should it be).”
Response: We agree with the reviewer and have changed statistical significance to precision.
Reviewer: “--Line 176: Clarify the meaning of “tended to” (I suggest avoiding this phrase, which does not have a precise statistical meaning).”
Response: We have changed “tended to” to “associated with” in the 3rd paragraph of the Results section.
Reviewer: “--Reword the following lines to avoid cause-effect claims such as “…sodium/creatinine ratio… increase…damage”: 176, 180.”
Response: We changed “increase” to “associated with an increase” in the 3rd paragraph of the Results section.
Reviewer--: “Use “observed effect(s)” or “estimated effect(s) instead of “effect(s)” in the following lines: 183, 87.”
Response: We changed the word “effect(s)” to “observed effects” in the 3rd and 4th paragraphs of the Results section.
Reviewer: “Tables--Indicate units for the beta-coefficients.”
Response: We have now added units for the beta-coefficients in the tables.
Reviewer: “Discussion--Reword the following lines to avoid cause-effect claims: 195, 225.”
Response: We have changed the reworded to “...associated with an increase...” in the 1st paragraph of the Discussion section.
Reviewer: “Use “observed effect(s)” or “estimated effect(s) instead of “effect(s)” in the following lines: 193, 195.”
Response: We strongly agree with the Reviewer’s suggestion and have changed the word “effect” to “observed effect” or “estimated effect” throughout the manuscript when referring to our results.
Reviewer: “Clarify the meaning of “tended to” (I suggest avoiding this phrase, which does not have a precise statistical meaning), in the following lines: 193, 224
Response: We replaced the wording “tended to increase” with “associated with an increase” in the 1st and 2nd paragraphs in the discussion section.
Reviewer: “Line 227: refer to precision rather than statistical significance, given that your analysis is not based on significance testing (nor should it be).”
Response: Again, we strongly agree with the Reviewer. We have now deleted the wording “statistical significance” throughout the manuscript and replaced it with the word “precision” in the 2nd paragraph of the Discussion section.
Reviewer: “Line 230: If this is the first use of ODC, spell it out and describe what it is here rather than later in paragraph.”
Response: We have now spelled out ODC where it first appears in the 3rd paragraph of the Discussion section.
Reviewer: “Line 231: “colonization of H. pylori” should be “H. pylori colonization” or “colonization by H. pylori.”
Response: We have now changed the wording to “H. pylori colonization” in the 3rd paragraph of the Discussion section.
Reviewer: “Line 246: I suggest changing “opinion” to “expert judgment.”
Response: We have changed the wording as suggested by the Reviewer in the 4th paragraph of the Discussion section.
Reviewer: “Lines 248-9: given that predictions about “bias” are generally relevant only to expected results, I suggest rewording this sentence to something like “If the non-differential misclassification was independent of other variable classification errors, it would be expected to result in underestimation of the magnitude of the association.’”
Response: We have made the suggested wording change in the 4th paragraph of the Discussion section.
Reviewer: “Lines 252-264: I suggest dropping the discussion of multiple hypothesis testing given the following: the description of statistical methods does not refer to statistical hypothesis testing as a goal; the analysis appropriately presents confidence intervals rather than p-values; the use of statistical significance testing and Bonferroni corrections have long been considered poor practice in epidemiology. Instead, I suggest restricting any justifications of the statistical methods used to the Methods section, where, as noted above, I suggest that you explicitly describe and justify the measures used for effect estimation. In that section, you can reference epidemiologic methods literature that explains why effect estimation, rather than statistical significance testing is the appropriate analytic goal. If you feel it’s necessary, you can also reference literature that explains why the Bonferroni correction is not appropriate.”
Response: We fully agree with the reviewer and have removed these lines from the discussion section. We have added references which justifies why measures used for effect estimation, rather than significance testing, is the analytic goal.
Reviewer: “Conclusion: Line 293: I suggest removing the first sentence of the conclusion, which is overly vague.”
Response: We have removed that sentence from the manuscript.
Reviewer: “Lines 298-300: Clarify the meaning of the last sentence of the conclusion; as written; it sounds like it is a foregone conclusion that salt will prevent progression of gastric cancer progression, that doing so is of relevance to all patients, that clinicians need help to advise patients to reduce salt intake, and that understanding mechanisms will help clinicians advise their patients, none of which can be assumed to be true.”
Response: We have now removed the previous last sentence of the conclusions. We believe that the placement of the sentence as the last sentence of the conclusion may have given the reader an impression that what we were saying was a foregone conclusion. We have now added a more general sentence to the beginning of the conclusions which is meant to provide the reader with some clarity as to why we conducted the present analyses and how it may contribute to the literature.
Reviewer 4 Report
The aim of the manuscript "The association between salt and the potential mediators of the gastric precancerous process" was to find a correlation between the amount of salt consumed and gastric inflammation, epithelial damage, gastric epithelial cell proliferation and H. pylori density during infection. The prevalence of H. pylori in the world exceeds over 60%, which makes it important to know the socioeconomic and biological factors modulating the frequency and course of H. pylori infections.
The article is written correctly, but contains many elements that need improvement:
a) Introduction:
- “Progression from a normal gastric mucosa to gastric cancer is believed to involve a series of steps in which chronic non-atrophic gastritis progresses into multifocal atrophic gastritis, intestinal metaplasia, dysplasia, and gastric cancer in a sequential manner.” à the course of gastric cancers is not necessarily related to intestinal metaplasia, according to the latest research and opinions it is a false positive dependence [Graham DY, Zou WY. Guilt by association: intestinal metaplasia does not progress to gastric cancer. Curr Opin Gastroenterol. 2018; 34(6): 458-464. doi: 10.1097/MOG.0000000000000472.]
- the description of the introduction is very sketchy and does not introduce the Reader in the subject of the article, it is reflected in quoting a dozen articles in one place, which is an unacceptable operation, e.g. „…” [7-26] and „…” [7-25, 27-39]
b) Materials and Methods:
- I believe that measuring the amount of salt intake through the respondents' questionnaire is a very imprecise way of obtaining data, many people may not have such information, because they do not prepare their own meals
- determination of gastric inflammation, epithelial damage or H. pylori density using a 0-3 scale (without the use of any units) is associated with a high risk of errors and variability and is therefore unreliable
c) Results:
- the results are described very briefly, in a way that prevents its understanding to people who do not deal with this subject (the text should be simplified, which does not mean short)
- the proportions are shaken - the whole results are focused on table 1, while the description of tables 2 and 3 is contained in one sentence “Salt intake, measured either as the self-reported frequency of adding salt to foods or as the amount of salt added to foods, showed effects on epithelial damage which were less precise compared to the estimates from urinary sodium/creatinine ratios, but the estimates were in the same direction (Tables 2 and 3).”
d) Discussion:
- the discussion is not exhaustive and about half of this section is devoted to the limitations of the article
- “In our study, salt intake during the first 5 months tended to independently increase the density of polymorphonuclear and/or mononuclear cells, the density of H. pylori infection, and epithelial damage in humans over time, although these associations did not reach statistical significance.” à the lack of statistical significance means the lack of correlation between these processes, despite the negative or positive trend of changes
e) References:
- the lack of possibility of polemics with the latest reports on gastric cancer in the context of H. pylori infections
Author Response
We would like to thank the reviewers for their comments and suggested edits. Below are the reviewers’ comments (in quotation marks) followed by our point-by-point response (in italics).
Reviewer: “The aim of the manuscript "The association between salt and the potential mediators of the gastric precancerous process" was to find a correlation between the amount of salt consumed and gastric inflammation, epithelial damage, gastric epithelial cell proliferation and H. pylori density during infection. The prevalence of H. pylori in the world exceeds over 60%, which makes it important to know the socioeconomic and biological factors modulating the frequency and course of H. pylori infections.
The article is written correctly, but contains many elements that need improvement:
a) Introduction:
- ‘Progression from a normal gastric mucosa to gastric cancer is believed to involve a series of steps in which chronic non-atrophic gastritis progresses into multifocal atrophic gastritis, intestinal metaplasia, dysplasia, and gastric cancer in a sequential manner.’ à the course of gastric cancers is not necessarily related to intestinal metaplasia, according to the latest research and opinions it is a false positive dependence [Graham DY, Zou WY. Guilt by association: intestinal metaplasia does not progress to gastric cancer. Curr Opin Gastroenterol. 2018; 34(6): 458-464. doi: 10.1097/MOG.0000000000000472.]”
Response: We find the Reviewers’ comment to be interesting, and in fact, we would like to thoroughly review the literature on this topic from an epidemiological perspective in the future. We find this comment more applicable to our previous manuscript which specifically estimated the effect of salt intake on progression to atrophic gastritis, intestinal metaplasia, and dysplasia or gastric cancer rather than the current manuscript where the outcomes of interest are H pylori density, inflammation and epithelial damage. In the current analysis, intestinal metaplasia is not an outcome of interest. Although we do not believe the controversy is highly relevant to the current analyses, we have removed the sentence mentioned by the Reviewer, so as not to distract from the focus of the current analysis.
Reviewer: “- the description of the introduction is very sketchy and does not introduce the Reader in the subject of the article, it is reflected in quoting a dozen articles in one place, which is an unacceptable operation, e.g. „…” [7-26] and „…” [7-25, 27-39]
Response: In the last sentence of the introduction we specifically state what the subject of the article is: “... the associations between salt intake and potential mediators of the gastric precancerous process (gastric inflammation, epithelial damage, density of H. pylori infection, and gastric epithelial cell proliferation).” Although we have conducted a thorough review of the literature on studies examining the effect of salt on gastric cancer, our study under review is not a review article, and therefore our thorough review is not presented here. Our intention for providing reference citations for all studies conducted on the topic of salt intake and gastric cancer was to emphasize that there are many several that have been conducted on this general topic, but none of these specifically estimated the association between salt intake and potential mediators of the gastric precancerous process (gastric inflammation, epithelial damage and density of H pylori infection.) I suppose, we could have made the statement that there are no other studies which have examined the associations we are focusing on within the literature relevant to salt’s intake on gastric cancer without including any reference citations. However, we wanted to be transparent and helpful to the readers by providing a list of references to assist the readers if they wish to independently examine these articles to verify this. We are confused why the Reviewer believes this is unacceptable. We are not clear what revision to the Reviewer is looking for.
Reviewer: “b) Materials and Methods: - I believe that measuring the amount of salt intake through the respondents' questionnaire is a very imprecise way of obtaining data, many people may not have such information, because they do not prepare their own meals.”
Response: We strongly agree with the reviewer. In fact, for this reason we focused our findings primarily on the more objective sodium/creatinine ratios. We have now mentioned the limitations that the Reviewer notes in section 4.2 and in the Discussion section.
Reviewer: “- determination of gastric inflammation, epithelial damage or H. pylori density using a 0-3 scale (without the use of any units) is associated with a high risk of errors and variability and is therefore unreliable.”
Response: We have now added this limitation in the 4th paragraph of the Discussion section.
Reviewer: “c) Results:- the results are described very briefly, in a way that prevents its understanding to people who do not deal with this subject (the text should be simplified, which does not mean short).”
Response: This comment was also mentioned by Reviewer 3. As described in our response to Reviewer 3, we have now revised the interpretation of the results for clarity.
Reviewer: “- the proportions are shaken - the whole results are focused on table 1, while the description of tables 2 and 3 is contained in one sentence ‘Salt intake, measured either as the self-reported frequency of adding salt to foods or as the amount of salt added to foods, showed effects on epithelial damage which were less precise compared to the estimates from urinary sodium/creatinine ratios, but the estimates were in the same direction (Tables 2 and 3).’”
Response: As Reviewer 4 previously mentioned, ‘... measuring the amount of salt intake through the respondents' questionnaire is a very imprecise way of obtaining data, many people may not have such information, because they do not prepare their own meals.’ We agree with the reviewer and it is for this reason that we focus primary on the more objective and accurate measure of salt intake, the sodium/creatinine ratio which was reported in Table 1. We further agree with the Reviewer and have removed Tables 2 and 3, and now only include them in Supplemental Material in Table S1 and Table S2.
Reviewer: “d) Discussion:- the discussion is not exhaustive and about half of this section is devoted to the limitations of the article.”
Response: In this study we looked at the effect of salt intake on gastric inflammation, epithelial damage, H. pylori density, and the interaction between these factors. Since there is limited research (only in animal studies and no previous epidemiologic studies that we know of) on this topic, our discussion is short. We do not know what other information the Reviewer would like to see in the Discussion section.
Reviewer- “In our study, salt intake during the first 5 months tended to independently increase the density of polymorphonuclear and/or mononuclear cells, the density of H. pylori infection, and epithelial damage in humans over time, although these associations did not reach statistical significance.’ The lack of statistical significance means the lack of correlation between these processes, despite the negative or positive trend of changes.”
Response: As mentioned by Reviewer 3, we should not have used the wording “statistical significance.” We strongly agree with Reviewer 3, and have totally deleted this wording throughout the manuscript. Please see our response to Reviewers 1 and 3. Consistent with Reviewer 3’s comments, we are following recommendations from the American Statistical Association and leading methodologists in epidemiology. Please refer to the American Statistical Association’s recommendation regarding p-values and statistical significance (Ronald L. Wasserstein & Nicole A. Lazar (2016) The ASA's Statement on p-Values: Context, Process, and Purpose, The American Statistician, 70:2, 129-133, DOI: 10.1080/00031305.2016.1154108). Highlighting the rarity and importance of this statement (this educational policy statement was one of the few made by the ASA over the last 150 years). As pointed out by the American Statistical Association in their Statement, “let us be clear. Nothing in the ASA statement is new. Statisticians and others have been sounding the alarm about these matters for decades, to little avail. We hoped that a statement from the world’s largest professional association of statisticians would open a fresh discussion and draw renewed and vigorous attention to changing the practice of science with regards to the use of statistical inference.” Additionally, the March, 2019 issue of Nature also addresses the importance of not using statistical significance. We have added two references ([54, 55] corresponding to the ASA statement and the March, 2019 article in Nature) to justify our analytic focus.
Reviewer: “e) References: - the lack of possibility of polemics with the latest reports on gastric cancer in the context of H. pylori infections.”
Response: We are not clear what change to the references the reviewer is requesting us to make here. We did discuss potential effect modification by Helicobacter pylori status.
Reviewer 5 Report
The manuscript “The association between salt and the potential mediators of the gastric precancerous process” describes that salt appears to increase epithelial damage in stomachs with more severe previous H. pylori induced chronic inflammation via investigating the effects of salt on gastric inflammation, epithelial damage, density of H. pylori infection, and gastric epithelial cell proliferation. These above potential mediators are measured using gastric biopsies as: a) the density of polymorphonuclear and mononuclear cells (gastric inflammation), b) mucus depletion (gastric epithelial damage), and c) the severity of H. pylori infection. Salt intake is measured with spot urine samples (using urinary sodium/creatinine ratios), self-reported frequency of adding salt to food, and as total added salt.
Salt intake and tobacco smoking have been reported to act as the harmfully environmental carcinogens for gastric adenocarcinoma, authors should provide the more important impact about the findings in this study.
It is suggested to provide the number of cases in the Materials and Methods, Table 1, Table 2 and Table 3 for easy comprehension in this study.
Author Response
We would like to thank the reviewers for their comments and suggested edits. Below are the reviewers’ comments (in quotation marks) followed by our point-by-point response (in italics).
Reviewer: “The manuscript “The association between salt and the potential mediators of the gastric precancerous process” describes that salt appears to increase epithelial damage in stomachs with more severe previous H. pylori induced chronic inflammation via investigating the effects of salt on gastric inflammation, epithelial damage, density of H. pylori infection, and gastric epithelial cell proliferation. These above potential mediators are measured using gastric biopsies as: a) the density of polymorphonuclear and mononuclear cells (gastric inflammation), b) mucus depletion (gastric epithelial damage), and c) the severity of H. pylori infection. Salt intake is measured with spot urine samples (using urinary sodium/creatinine ratios), self-reported frequency of adding salt to food, and as total added salt.
Salt intake and tobacco smoking have been reported to act as the harmfully environmental carcinogens for gastric adenocarcinoma, authors should provide the more important impact about the findings in this study.”
Response: Consistent with suggestions by Reviewer 3, we do not want to overstate our conclusions, and rather we recommend further research in humans before more definitive conclusions be made on the impact that these findings may have.
Reviewer: “It is suggested to provide the number of cases in the Materials and Methods, Table 1, Table 2 and Table 3 for easy comprehension in this study.”
Response: We have now added flowcharts as supplemental materials for the sample size derivation and we have added sample sizes in the Table.
Round 2
Reviewer 1 Report
The authors made a thorough revision on the manuscript and I have no other comments.
Reviewer 4 Report
The authors of the article have adapted to the majority of Reviewers' suggestions. For this reason, the article can be published in its current form.